

# Daytime aerosol optical depth above low-level clouds is similar to that in adjacent clear skies at the same heights: airborne observation above the southeast Atlantic

Yohei Shinozuka[1,2], Meloë S. Kacenelenbogen[2], Sharon P. Burton[3], Steven G. Howell[4], Paquita Zuidema[5], Richard A. Ferrare[3], Samuel E. LeBlanc[2,6], Kristina Pistone[2,6], Stephen Broccardo[1,2], Jens Redemann[7], K. Sebastian Schmidt[8], Sabrina P. Cochrane[8,9], Marta Fenn[3,10], Steffen Freitag[4], Amie Dobracki[4,5], Michal Segal-Rosenheimer[2,6,11], Connor J. Flynn[7]

[1] Universities Space Research Association, Columbia, Maryland, USA

[2] NASA Ames Research Center, Moffett Field, California, USA

[3] NASA Langley Research Center, Hampton, Virginia, USA

[4] University of Hawaii at Manoa, Honolulu, Hawaii, USA

[5] University of Miami, Miami, Florida, USA

[6] Bay Area Environmental Research Institute, Moffett Field, California, USA

[7] University of Oklahoma, Norman, Oklahoma, USA

[8] University of Colorado, Boulder, Colorado, USA

[9] Laboratory for Atmospheric and Space Physics, Boulder, Colorado, USA

[10] Science Systems and Applications, Inc, Hampton, VA, USA

[11] Department of Geophysics, Porter School of the Environment and Earth Sciences, Tel-Aviv University, Tel-Aviv, Israel

*Correspondence to*: Y. Shinozuka (Yohei.Shinozuka@nasa.gov)





**Abstract**
To help satellite retrieval of aerosols and studies of their radiative effects, we demonstrate
that daytime 532 nm aerosol optical depth over low-level clouds is similar to that in neighboring
clear skies at the same heights in recent airborne lidar and sunphotometer observations above the
southeast Atlantic. The mean AOD difference is between 0 and -0.01, when comparing the cloudy
and clear sides, each up to 20 km wide, of cloud edges. The difference is not statistically significant
according to a paired t-test. Systematic differences in the wavelength dependence of AOD and in
situ single scattering albedo are also minute. These results hold regardless of the vertical distance
between cloud top and aerosol layer bottom. AOD aggregated over ~2º grid boxes for each of
September 2016, August 2017 and October 2018 also shows little correlation with the presence of
low-level clouds. We posit that a satellite retrieval artifact is entirely responsible for a previous
finding of generally smaller AOD over clouds (Chung et al., 2016), at least for the region and time
of our study. Our results also suggest that the same values can be assumed for the intensive
properties of free-tropospheric biomass-burning aerosol regardless of whether clouds exist below.
**1.      Introduction**
A significant amount of atmospheric particles are transported above liquid water clouds on
the global scale (Waquet et al., 2013). Aerosols above clouds (AAC) may influence the climate in
three ways. Their light absorption is amplified by cloud reflection. The heating of the atmosphere
due to the absorption may stabilize the atmosphere. The particles may eventually subside, enter
clouds and alter their properties. Estimates of the direct aerosol radiative effect alone see large
inter-model spread for areas with large aerosol optical depth (AOD) over widespread clouds (Stier
et al., 2013; Zuidema et al., 2016).
Since AAC are difficult to see from the ground or a ship, previous studies have relied on
satellite observations (see Table 2 of Kacenelenbogen et al., 2019). Among them is Chung et al.
(2016), which used the level 2 products of the Cloud-Aerosol Lidar with Orthogonal Polarization
(CALIOP) (Winker et al., 2009) to calculate the AOD above the maximum low-cloud-top-height
in each grid cell in clear sky as well as the AOD above low clouds on a global 2° × 5° latitude–
longitude grid. Their results indicate that daytime 532 nm AOD above low clouds is generally



lower than that in clear sky at the same heights. The difference is up to 0.04 over the southeastern
Atlantic Ocean (see their Fig. 2)

As Chung et al. (2016) point out, it is conceivable that aerosol amounts over cloud can be

different from those in nearby clear sky. There are two sets of potential reasons. The first concerns
the effects of meteorology. Large-scale circulation patterns paired with solar reflection from clouds
on aerosols could modify the horizontal and vertical extent of aerosols, aerosol concentration and
chemical composition. For example, the properties of hygroscopic aerosols might vary if the
relative humidity in clear skies is somehow higher than above clouds. The second set of reasons
pertain to the case of aerosols in close proximity to clouds. The proximity has been variously
defined, for example less than 100 m in the vertical direction (Costantino and Bréon, 2013) and
less than 20 km in the horizontal direction (Várnai and Marshak, 2018). Chung et al. (2016) note
that aerosols were shown to influence underlying cloud by indirect effects and semidirect effects
(Costantino and Bréon, 2010, 2013; Johnson et al., 2004; Wilcox, 2010) and that these aerosol–
cloud interactions and possibly more (e.g., a pronounced if unlikely aerosol entrainment (Diamond
et al., 2018)) might somehow affect the aerosol amount over cloud. A bias in the CALIOP standard
retrieval was also raised as a possible explanation for the Chung et al. (2016) results. The detection
threshold in the feature detection algorithm varies depending on the background lighting
conditions, the atmospheric features (e.g., aerosols, high altitude cirrus or boundary layer clouds)
and the horizontal averaging required by CALIOP for detection (see Fig. 4 of Winker et al. (2009)).
In the particular case of aerosols above clouds, Kacenelenbogen et al. (2014) show that the
CALIOP standard algorithm substantially underestimates the frequency of AAC when the AOD
is less than ~0.02. This is due mostly to tenuous aerosols with a backscatter under the detection
threshold; however, Kacenelenbogen et al. (2014) saw no clear bias in AOD above clouds between
CALIOP and the NASA Langley airborne High Spectral Resolution Lidar (HSRL-1). Liu et al.
(2015) show a clear AOD underestimate of the CALIOP level 2 retrieval in comparison to a
separate retrieval after Hu et al. (2007) for smoke above opaque water clouds over the southeast
Atlantic, and explain this by the CALIOP layer detection scheme prematurely assigning layer base
altitudes and thus underestimating the geometric thickness of smoke layers. According to Chung
et al. (2016), the negative daytime AOD differences between cloudy and cloud-free conditions
"might simply be a result of systematic differences between the detection thresholds in clear sky



and above low bright clouds". The authors add that the bias may be enhanced over the ocean due
to the lower albedo compared to that of land.
The subject warrants further investigation, given the importance of AAC on climate. An
airborne experiment can help by providing direct measurements that are subject to smaller
uncertainty, in finer spatial and temporal resolution albeit over limited ranges. The NASA
ObseRvations of Aerosols above CLouds and their intEractionS (ORACLES) mission was carried
out to study key processes that determine the climate impacts of African biomass-burning aerosols
above the southeast Atlantic. Of the two deployed aircraft, the NASA P3, equipped with in situ
and remote sensing instruments, flew in the lower- to mid-troposphere, mostly in September 2016,
August 2017 and October 2018. In September 2016 the NASA ER2 also flew, at about 20 km
altitude with downward-viewing sensors. Extensive stratocumulus clouds were observed
repeatedly throughout the mission; see a sample satellite image in Redemann et al. (in preparation).
Details of the ORACLES mission can be found in Redemann et al. (in preparation), Zuidema et
al. (2016) and Shinozuka et al. (2019).
The instrumentation relevant to the present paper is described in Sect. 2 along with
sampling and statistical hypothesis testing methods. This is followed by comparisons of AOD and
other aerosols properties above the height of cloud top between cloudy and clear skies (Sect. 3).
Sect. 4 offers discussion.
**2. Methods**
**2.1. Instrumentation**
The remote sensing and in situ instruments used in this study are briefly described below
with references to full descriptions. Note that the measurements each refer to a unique vertical
range, as summarized in Table 1.
The NASA Langley Research Center High Spectral Resolution Lidar (HSRL-2), deployed
from the ER2 in 2016 and from the P3 in 2017 and 2018, measures calibrated, unattenuated
backscatter and aerosol extinction profiles below the instrument. The data are reported with 10 s
intervals. The HSRL-2 signal-to-noise ratio is higher than that of CALIOP, due to the much lower
altitude and the inverse square dependence of light intensity. In addition, by the use of a second
channel to assess aerosol attenuation, the HSRL technique (Shipley et al., 1983) results in an
accurate aerosol extinction product with no assumptions about lidar ratio, and also a more accurate



backscatter product, particularly in the lower atmosphere where attenuation by upper layers can
present difficulties for the spaceborne backscatter lidar. Differences in algorithm are discussed in
Sect. 4. Further details about the instrument, calibration and uncertainty can be found in Hair et al.
(2008), Rogers et al. (2009) and Burton et al. (2018).
Our analysis utilizes the HSRL-2 standard products of cloud top height (CTH), 532 nm
particulate backscattering and 532 nm aerosol optical thickness (Burton et al., 2012) in three ways.
First, flight segments are isolated using the CTH product (detailed in Sect. 2.2). Second, the bottom
and top heights of the smoke plumes are defined with a (somewhat arbitrarily chosen) threshold
backscattering coefficient at 0.25 $Mm^{-1}sr^{-1}$ after Shinozuka et al. (2019).
Third, we evaluate the 532 nm partial-column aerosol optical thickness from below the
aircraft down to ~50 m above the CTH, (even for columns without clouds; see Sect. 2.2). The ~50-
m buffer is designed to reduce the ambiguity associated with the transition at the cloud top. The
upper limit of the integral of extinction is 14 km altitude for the 2016 ER2 flights and 1500 m
below the P3 altitude for 2017 and 2018. Profiles with possible influences of mid- and high-level
clouds are largely excluded from the product, though isolated cases of thin clouds may remain.
We also use partial-column AOD observed upward from the P3 with a sunphotometer. The
Spectrometer for Sky-Scanning, Sun-Tracking Atmospheric Research (4STAR) measures hyper-
spectral direct solar beam. Calculated AOD is reported at 1 Hz. Our analysis excludes data with
possible influences of clouds above the instrument. Further details on the instrument as well as
data acquisition, screening, calibration and reduction can be found in Dunagan et al. (2013),
Shinozuka et al. (2013) and LeBlanc et al. (2019).
For 2017 and 2018, we examine a combination of the 4STAR and HSRL-2 AODs, in order
to cover the free troposphere both upward and downward from the aircraft that flew in it (Fig. 1).
The vertical coverage is compromised by two limitations intrinsic to the lidar measurements. First,
the CTH is not sought within 500 m of the instrument (not to be confused with the ~50-m lower
buffer for the extinction integral). This means that the flight segments with clouds so close to the
aircraft enter our analysis only if the clouds extended as deep as to reach 500 m away from it. This
is at most a minor fraction of the data, as the fraction with the CTH within 550 m of the P3 altitude
is a mere 3%. Second, because of the 1500 m upper buffer for the extinction integral, we only have
4STAR above-P3 AOD for the flight segments when the plane was 500-1500 m above the CTH





(Fig. 1b). We add the HSRL-2 AOD to the 4STAR AOD only for the flight segments when the P3
was >1500 m above the CTH (Fig. 1c).
For 2016, we examine the HSRL-2 AOD only, because, with the lidar above the
troposphere, two of the missing layers can safely be ignored, leaving the ~50 m lower buffer as
the only missing layer (Fig. 1a). We refer to all these AODs from the three campaigns collectively
as $AOD_{ct}$ (see Table 1). The wavelength dependence expressed as Angstrom exponent is calculated
for 10-s periods with $AOD_{ct}$ at 355 and 532 nm both exceeding 0.1.
In situ aerosol instruments operated from the P3 include a nephelometer (TSI model 3563)
and a particle soot absorption photometer (PSAP, Radiance Research 3-wavelength version),
which measure particulate light scattering and absorption, respectively. After adjustments are
made for factors such as angular truncations (Anderson and Ogren, 1998) and filter interference
(Virkkula, 2010) for each wavelength, extinction coefficient and single scattering albedo at 550
nm are derived for an instrument relative humidity (RH) that is typically below 40%. Pistone et al.
(2019) and Shinozuka et al. (2019) have more details. The non-refractory masses of submicron
particles were measured by a time-of-flight aerosol mass spectrometer (Aerodyne, Inc HR-ToF
AMS, DeCarlo et al. (2006)). A condensation particle counter (TSI model 3010, with ΔT set to
22°C) measured the number concentration of particles larger than about 10 nm. These in situ
properties refer to the air immediately outside the P3 aircraft, not a vertical column. Only the in
situ measurements in 2017 and 2018 at 500-1500 m above the CTH are used in this study.
**2.2.    Sampling**
Two methods are employed for selecting subsets of the observations for analysis. In the
first (Sect. 2.2.1), we bundle data from areas hundreds of kilometers wide for each of the three
campaigns, in a manner as similar to the CALIOP-based study (Chung et al., 2016) as the airborne
measurements allow. In the second method (Sect. 2.2.2), we pair cloudy and clear skies with more
stringent spatiotemporal criteria to isolate the impact of finer-scale phenomena. Note that both
methods ignore time periods for which the 532 nm backscattering product (from which the CTH
product is derived) is masked at all altitudes, as well as transit flights into and out of the study area.
Cases are also excluded where the CTH exceeds 3241 m. This is to be consistent with the study
by Chung et al. (2016), which refers to clouds at 680 hPa or higher pressure, although we find
similar results with or without this restriction.



### 2.2.1. Meso-scale monthly-mean sampling


This method separates profiles measured in the three campaigns into two groups: those
concurrent with a presence of low-level clouds as reported by the HSRL-2 and those concurrent
with an absence of any cloud detected by HSRL-2 in the column. The groups are each aggregated
into grid boxes approximately 2° by 2°, as shown in Fig. 2. This grid is adapted from Shinozuka et
al. (2019) but with additional boxes for the São Tomé-based 2017 and 2018 campaigns. In total,
109 hours and 39 hours of flight segments are selected for the cloudy and clear groups,
respectively, including minor double-counting where boxes overlap.
The arithmetic mean of the CTH of the cloudy group is calculated for each day for each
box and 50 m above it is set as the lowest altitude for computing $AOD_{ct}$ for each 10 s period (Sect.
2.1). Then the arithmetic mean and standard deviation are calculated for the $AOD_{ct}$, as well as
other measurements (Sect. 2.1, Table 1), for each group and each box. After excluding the time
periods with mid- and high-level clouds and instrument/aircraft issues, 49 hours and 26 hours of
the $AOD_{ct}$ measurements enter the analysis for cloudy and clear-sky groups, respectively.

### 2.2.2. Local-scale near-synchronous sampling


This method identifies cloud edges and demarcates the cloudy side and clear side of each
edge based on the time series of the CTH detected by HSRL-2, for level flight legs only. Cloud
edges are defined by the points in time when a cloud is detected in a profile adjacent to a profile
with no cloud detection.
A clear sky and a cloud are represented by the time period of a certain length, say 60 s,
preceding each edge and the same length following it. To ensure that clear skies and clouds are
not interrupted for the length, we exclude edges for which another one is found within the length.
The longer the length, the smaller the number of cloudy-clear pairs, because longer continuous
clouds and clear skies are rarer. Furthermore, we set another length, 20 s in the example illustrated
in Fig. 3a, to exclude immediately before and after the edge, in order to reduce ambiguity
associated with a gradual transition from cloud droplets to unactivated particles, the so-called
twilight zone (Koren et al., 2007; Schwarz et al., 2017; Várnai and Marshak, 2018). We convert
the temporal dimensions into horizontal ones using the mean true horizontal aircraft speed, 200
ms$^{-1}$ for the ER2 (Fig. 3a) and 140 ms$^{-1}$ for the P3 (Fig. 3b and Fig. 3c).





We change both the maximum and minimum limits of separation, in order to assess scale
dependence and sampling error as much as our airborne data permit. The way the edges are
identified ensures that a measurement cannot be counted more than twice for a given range of
separation. A measurement can, however, enter multiple ranges of separation. For example, a
measurement 4-6 km away from a cloud edge enters the ranges of 0-6 km, 2-6 km, 2-12 km, 4-12
km, 4-24 km, etc. In total, 5.0 hours of horizontal flight are selected, including the double-counting
for a given range but excluding the multiple-counting over multiple ranges. Exactly half of them
are over clouds. Note that these expressions of separation are only notional; we discuss this in Sect.
4.

As with the meso-scale monthly-mean sampling, we take the arithmetic mean of the CTH
of the cloudy side and add 50 m (red lines in Fig. 3). The height is extended to the adjacent clear
sky (orange lines) for the calculation of $AOD_{ct}$ (Sect. 2.1). The in situ measurements (Sect. 2.1,
Table 1) are each averaged over the cloudy sides and over the clear sides. Cases where aerosol
measurements are unavailable for 33% or more of the time period, for example due to calibration
or operation problems, are excluded. This makes the number of cloudy-clear pairs vary from
property to property for a given range of separation. In total, 3.8 hours of $AOD_{ct}$ measurements
enter the analysis.
**2.3.   Statistical hypothesis testing**
We employ the paired t-test, also called paired-samples t-test or dependent t-test, to
determine whether the mean difference in each property (e.g., $AOD_{ct}$) between the presence and
absence of low-level clouds is statistically consistent with the null hypothesis of zero difference.
The procedure entails calculating the t statistic, the ratio of the mean cloudy-clear differences to
their standard error. Here the standard error is the standard deviation computed for N-1 degrees of
freedom divided by the square root of N, where N is the number of sample pairs. Note that the
standard deviation is close to the root-mean-square deviation (RMSD) for small absolute mean
difference, unless N is smaller than five.
For the calculated t statistic, the two-tailed p value is looked up. Small p values are
associated with large t statistics and hence generally large mean differences relative to RMSD. If
the p value is smaller than 0.05, we reject the null hypothesis. If it is greater, we do not.





The procedure makes several assumptions. One is independence of the differences.
Synoptic- and meso-scale phenomena prevalent throughout ORACLES (e.g., subsidence and
anticyclones) reduce the independence of the samples. The low day-to-day meteorological
variability and repeated flight paths might mean that the same aerosol-cloud conditions were
sampled day after day. It is unclear whether this would reduce the independence of the cloudy-
clear differences - a potential, seemingly untestable caveat for the meso-scale monthly-mean
sampling (Sect. 2.2.1). In the local scale the exclusion of contiguous cloud edges (Sect. 2.2.2)
should attain a high level of independence from one another. The procedure also assumes
continuous (not discrete), approximately normally distributed data free of outliers.
**3.    Results**
The meso-scale monthly-mean method finds little systematic difference in 532 nm $AOD_{ct}$
(Fig. 4). Most markers lie near the 1:1 line. The mean difference, an indicator of systematic
differences, is +0.02. This is only +16% of the RMSD, an indicator of the total (random and
systematic) variability. The p value from the paired t-test is 0.23. Thus, the AOD above low-level
clouds is not significantly different from that at the same heights above nearby clear skies in this
scale. The p value is also greater than 0.05 for $\log_{10}$ of $AOD_{ct}$, the Angstrom exponent and in situ
aerosol properties (Table 2, see the rows labeled "box means").
The only exception is the particle number concentration. Four of the 32 horizontal boxes
see 3-7 times as large concentration over clouds as that over neighboring clear skies (4600-5700
$cm^{-3}$ vs. 700-2100 $cm^{-3}$). The mean cloudy-clear difference among all box means is about +40%
of the RMSD. The t-test yields a p value of 0.01. One of the assumptions underlying the test, the
absence of outliers, may be broken in this case.
The local-scale near-synchronous method finds virtually the same results. The $AOD_{ct}$ is
compared in Fig. 5a for 2-6 km separation. The time period corresponds to approximately 10-30 s
temporal range on the ER2 (13 data points from the 2016 campaign) and 14-43 s at the average P3
speed (53 from 2017 and 2018). All data points lie near the 1:1 line. The mean difference, -0.002,
is only -21% of the RMSD for 2-6 km separation. The p value is 0.08.
We run the same calculation for other combinations of minimum and maximum separation.
Fig. 6 shows the resulting statistics. The mean difference for 2-6 km separation, for example, is
represented in Fig. 6a at maximum separation (x axis) of 6 km by the solid orange line that starts





after the minimum separation of 2 km. This line also shows that the mean difference is -0.01 if the
maximum separation is set to 20 km while keeping the minimum at 2 km. The longest blue line
represents the calculations for zero minimum separation (i.e., with the twilight zone included). All
other solid lines represent the results with greater minimum separation. For example, the green
line that is missing data up to 4 km indicates that the mean difference is closer to -0.01 at 12 km,
as shown in Fig. 5b.

For the separation up to 20 km, the mean difference is mostly between 0 and -0.01. The p

value, shown in Fig. 6b, is below 0.05 for only a handful of the ranges of separation, many with
minimum separation of 0-2 km. This is also true for $\log_{10}$ of $AOD_{ct}$, the Angstrom exponent and
in situ aerosol properties including the number concentration (Table 2). Large p values are also
found for the ER2- and P3-borne measurements separately and for the 4STAR and the HSRL-2
AOD separately for 2017 and 2018.
**4.    Discussion and Conclusions**

Virtually no systematic differences in aerosol properties are found between the air above

low-level clouds and that above clear areas nearby in ORACLES daytime airborne measurements.
The finding holds for a range (0-20 km) of distances between, and expanses of, the two air masses.
Note that the temporal and horizontal dimensions associated with the local-scale near-synchronous
sampling must be collectively overestimated, because the aircraft may have been running parallel
to cloud edge. There is no easy way to know how far from the nearest cloud edge the airplane was
in reality. Images from cameras on the plane and satellites may give some context. But we stop
short of examining them, discouraged by the perceived difficulty in unifying definition of cloud
edges between the cameras and the lidar, among other image processing issues. Although we do
not know what the real distances and expanses are, that probably does not matter for the region
and season of our study, judging by the consistently large p values across the notional distances
and expanses. The meso-scale monthly-average sampling, resting on larger data, provides
consistent results. Note that this conclusion may or may not apply to environment elsewhere with
less uniform clouds.

Our analysis does not support aerosol-cloud interactions, circulation patterns or anything

else as a cause for a significant systematic difference, simply because such a difference is not
evident. The lack of obvious sensitivity to the smoke-cloud gap height, indicated by marker color



in Fig. 5, is consistent with this conclusion. The smoke bottom height minus the mean CTH gives
an estimate of whether aerosols may be physically in contact with clouds and therefore there is a
chance of wet removal. Our analysis does not detect any sign of local aerosol removal by the
underlying clouds.

An important difference between the present analysis and the CALIOP-based one (Chung

et al., 2016), apart from the spatiotemporal range and resolution, is that the HSRL algorithm (Hair
et al., 2008) does not use any explicit layer detection. The return signal in the molecular signal
provides a measure of the aerosol attenuation and extinction. A very tenuous aerosol layer still
produces a reported extinction with a reported error bar. If the aerosol extinction is very small, the
error bar may exceed the retrieved value, but there is no cutoff at small values that produces the
kind of bias one gets from a detection threshold. Furthermore, the signal-to-noise is higher than
that of CALIOP, as explained in Sect. 2.1.

We posit that the systematic differences shown in Chung et al. (2016) are solely a CALIOP

retrieval artifact, at least for the ORACLES region and season. As the authors discuss, the CALIOP
standard algorithm has a detection bias. The algorithm confines itself to distinct aerosol layers
whose signals are high enough compared to detector noise and, during the day, solar background
light. If the signal-to-noise ratio of a layer is not high enough, no extinction is reported for the
portion of the aerosol profile; summing up the extinction produces a low-biased AOD.

The depolarization/multiple scattering method by Hu et al. (2007) retrieves above-cloud

AOD from CALIOP without a layer detection algorithm. This method may lead to a different result
from Chung et al. (2016). A future study based on the Hu et al. method and extended to the globe
as in Kacenelenbogen et al. (2019) will also address environment under a wider variety of synoptic-
and meso-scale conditions that produce specific opaque water clouds.

The absence of systematic differences is good news, because satellite retrievals and studies

of radiative effects do not need to treat these two conditions as different. Our results on $AOD_{ct}$
justify, for example, temporal and horizontal extrapolation of above-cloud AOD to adjacent clear
skies and attribution of the difference from full-column AOD to the planetary boundary layer. Our
results on the aerosol intensive properties suggest that a single set of aerosol models can be used
for the aerosols in the free troposphere regardless of whether clouds exist below, which may allow
better characterization of the underlying clouds and the radiative effects (Matus et al., 2015; Meyer
et al., 2015). It seems reasonable to use aerosol properties retrieved in clear skies for estimating



the direct radiative effects of aerosols above nearby clouds. But challenges remain. Random
variability in AOD and other aerosol properties is significant, as indicated by RMSD in the present
study and quantified for smoke elsewhere (Shinozuka and Redemann, 2011). It may be
problematic to assume the same values for intensive properties for reasons not investigated here,
for example: form of combustion, degree of aerosol ageing and influence of the boundary layer.
These may be tackled more effectively by combining sensors of various capabilities with improved
spatiotemporal resolution and retrieval algorithms (National Academies of Sciences, Engineering,
and Medicine et al., 2019). Improved spatiotemporal satellite observations of aerosol properties in
clear skies and above clouds are urgently needed to reduce the uncertainty in total aerosol radiative
forcing (National Academies of Sciences, Engineering, and Medicine et al., 2019). For this, we
are looking forward to the next generation of space-borne lidars, radars, microwave radiometers,
polarimeters and spectrometers such as the ones that will address joint Aerosols, Clouds,
Convection and Precipitation (ACCP) science goals and objectives (https://science.nasa.gov/earth-
science/decadal-accp)
**Data availability**
The P3 and ER2 observational data (ORACLES Science Team, 2017, 2019) are available
through www.espo.nasa.gov/oracles.
**Author contribution**
All authors participated in the investigation during the ORACLES intensive observation
periods. In addition, MSK led conceptualization, funding acquisition, methodology, project
administration and supervision. YS led data curation, formal analysis, software and validation and
wrote the original draft. YS and MSK contributed visualization. All but CJF reviewed and edited
the manuscript.
**Competing interests**
The authors declare that they have no conflict of interest.



**Acknowledgments**
We thank Eric Wilcox, Tamás Várnai and Sasha Marshak for discussion. ORACLES is
funded by NASA Earth Venture Suborbital-2 grant NNH13ZDA001N-EVS2.

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





**Table 1. Properties and instruments used in this study and the altitudes they refer to.**

| Property | September 2016 on the ER2 aircraft | | August 2017 and October 2018* on the P3 aircraft | |
|---|---|---|---|---|
| | Instrument | Altitude | Instrument | Altitude |
| cloud top height (CTH) | HSRL-2 | limited to ≤3241 m in this study | HSRL-2 | no higher than 500 m below the P3 and ≤3241 m |
| aerosol optical depth above cloud top height (AOD$_{ct}$) | HSRL-2 | from ~50 m above the CTH to 14 km | 4STAR | from the P3 to top of atmosphere (TOA), when the P3 is 500-1500 m above CTH |
| | | | HSRL-2 and 4STAR | from ~50 m above the CTH to TOA, except 0-1500 m below the P3, when the P3 is >1500 m above CTH |
| extinction coefficient, single scattering albedo, submicron non-refractory organic mass, number concentration | - | - | nephelometer, PSAP, HR-ToF AMS and condensation particle counter | at the P3 when the P3 is 500-1500 m above CTH |

* One day in September 2017 and two days in September 2018 are also included.
- Not presented in this study. Observations were made from the P3, away from the ER2 for most
cases.







**Table 2. Statistics on the cloudy-clear differences**

| Sampling* | Mean Difference | RMSD | p | Number of Pairs |
|---|---|---|---|---|
| 532 nm $AOD_{ct}$ | | | | |
| 2-6 km | -0.00 | 0.01 | 0.08 | 66 |
| 4-12 km | -0.01 | 0.02 | 0.23 | 18 |
| box means | +0.02 | 0.12 | 0.23 | 54 |
| $\log_{10}$ 532 nm $AOD_{ct}$ | | | | |
| 2-6 km | -0.00 | 0.01 | 0.15 | 66 |
| 4-12 km | -0.00 | 0.02 | 0.27 | 18 |
| box means | +0.05 | 0.19 | 0.07 | 54 |
| Angstrom Exponent of $AOD_{ct}$ | | | | |
| 2-6 km | -0.04 | 0.11 | 0.00 | 53 |
| 4-12 km | -0.02 | 0.05 | 0.08 | 16 |
| box means | -0.02 | 0.10 | 0.14 | 54 |
| In Situ 550 nm Extinction Coefficient ($Mm^{-1}$) | | | | |
| 2-6 km | -0.2 | 3.0 | 0.87 | 7 |
| 4-12 km | -3.6 | 5.1 | 0.31 | 3 |
| box means | +19.0 | 67.9 | 0.18 | 24 |
| In Situ 550 nm Single Scattering Albedo | | | | |
| 2-6 km | -0.00 | 0.01 | 0.14 | 7 |
| 4-12 km | -0.01 | 0.01 | 0.35 | 3 |
| box means | -0.00 | 0.05 | 0.99 | 24 |
| Submicron Non-refractory Aerosol Organic Mass ($\mu g m^{-3}$) | | | | |





| 2-6 km | +0.1 | 0.5 | 0.75 | 9 |
|---|---|---|---|---|
| 4-12 km | -0.4 | 0.6 | 0.38 | 3 |
| box means | +1.2 | 4.4 | 0.14 | 28 |
| Number Concentration of Particles >10 nm ($cm^{-3}$) | | | | |
| 2-6 km | 5 | 59 | 0.82 | 10 |
| 4-12 km | -110 | 121 | 0.09 | 3 |
| box means | 614 | 1411 | 0.01 | 31 |

* Either the temporal separation from cloud edges or box means.



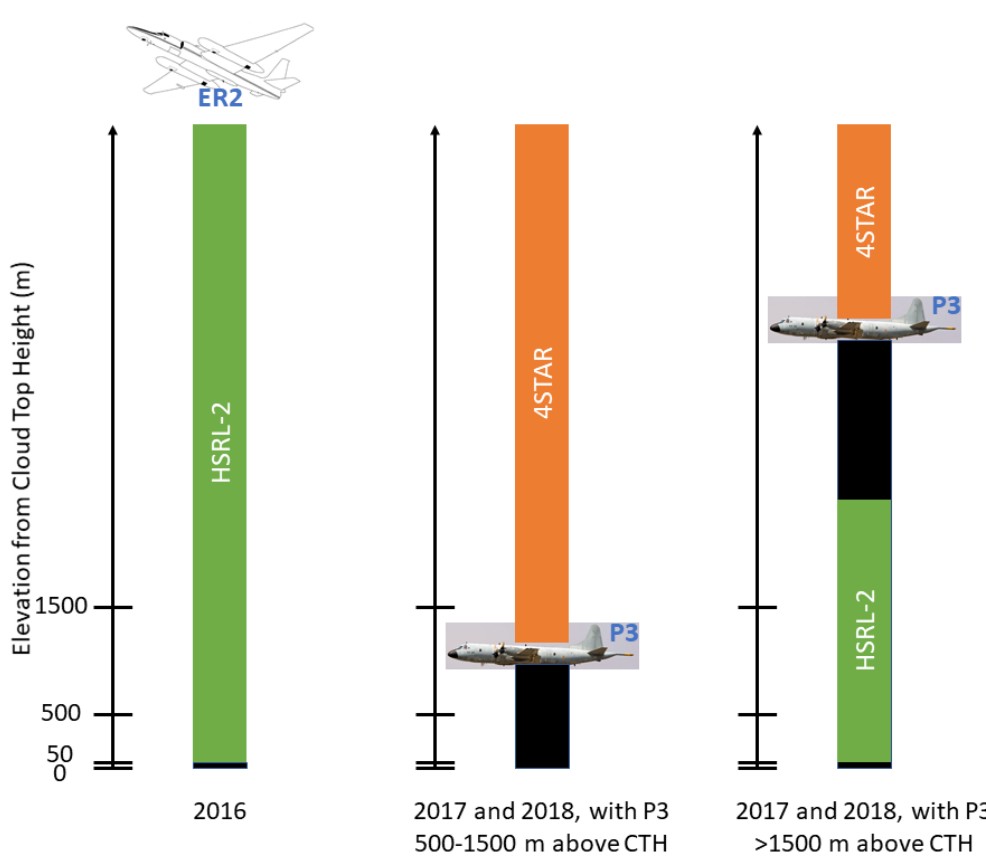


**Figure 1. AOD above cloud top height (AOD$_{ct}$). See text and Table 1 for details.**







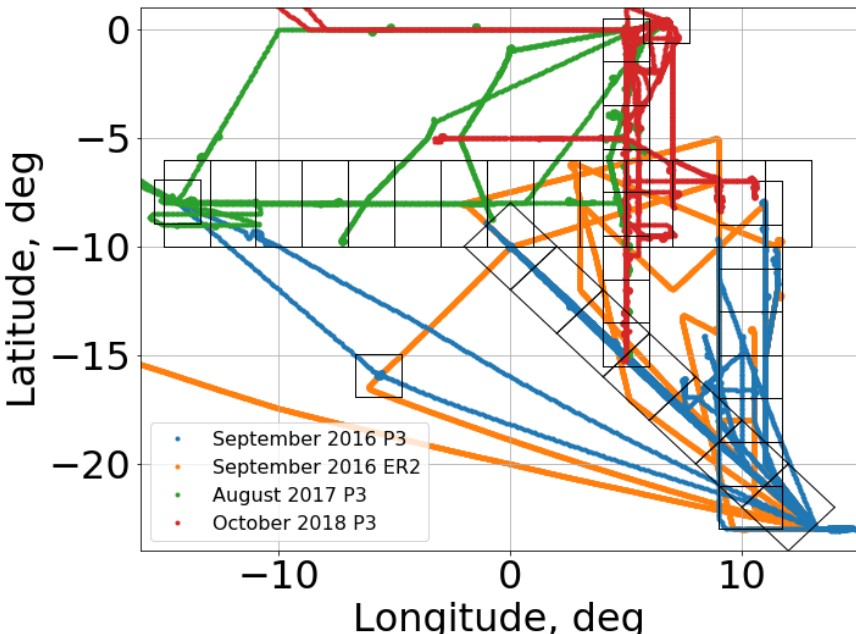


**Figure 2. The flight paths of ORACLES. The boxes for meso-scale monthly-mean sampling are superimposed.**




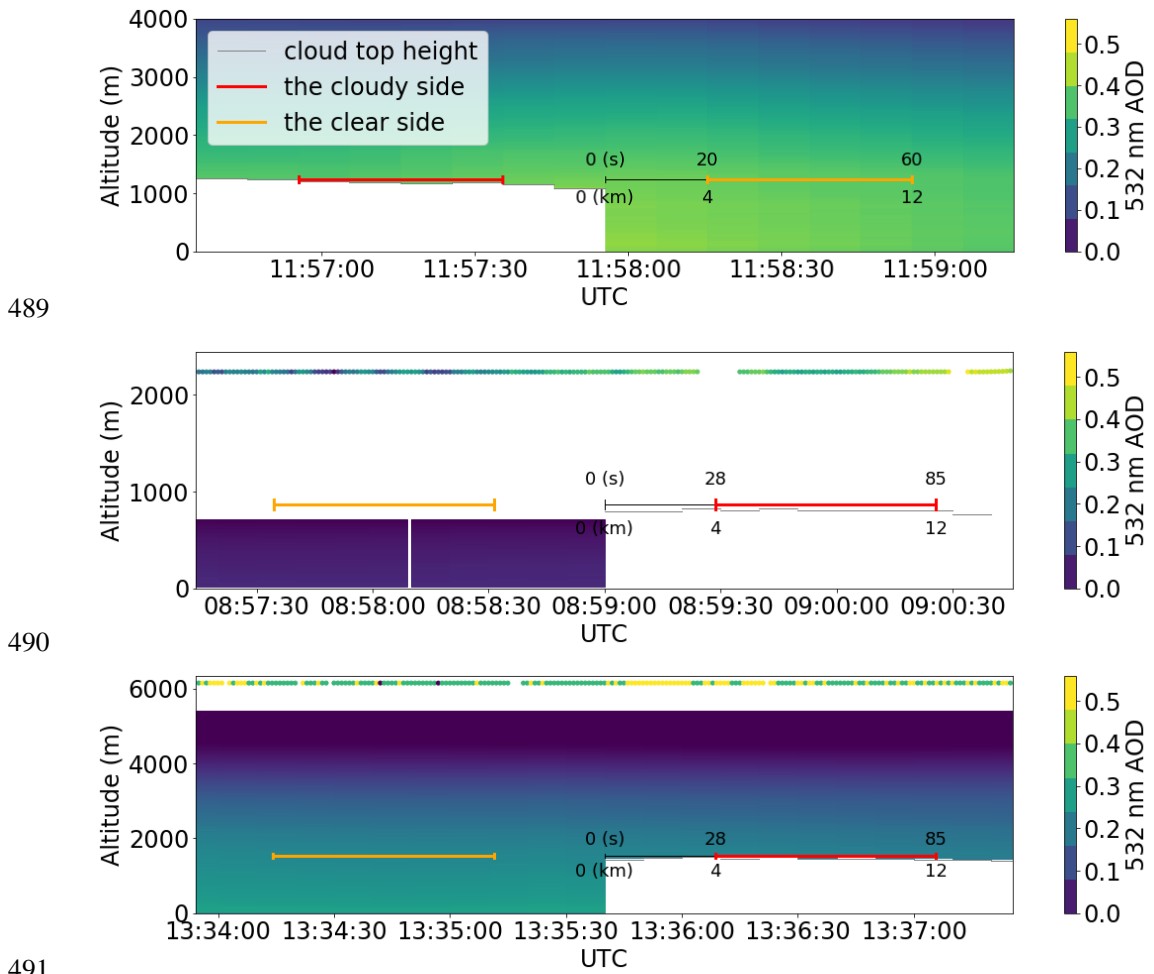




**Figure 3. (a) Local-scale near-synchronous sampling based on the HSRL-2 cloud top height (CTH) product. In this subset of the ER2 flight on September 12, 2016, a cloud edge is found at 11:57:56. The cloudy and clear side, each with horizontal separation of 4-12 km measured from cloud edge, are marked by red and orange lines, respectively. The HSRL-2 AOD profiles are given for altitudes from ~50 m above the CTH. (b) An example of local-scale near-synchronous sampling from the P3 aircraft with both HSRL-2 and 4STAR onboard. With the P3 500-1500 m above the CTH, as is the case with this example from October 5, 2018, we use 4STAR AOD only. The 4STAR AOD is indicated at the P3 altitudes just above 2000 m but refers to all altitudes above them. (c) Another example of local-scale near-synchronous sampling from the P3 aircraft with both HSRL-2 and 4STAR onboard. For the time periods when it flew >1500 m above the CTH, as is the case for this example from**





**October 15, 2018, the 4STAR AOD, indicated at the P3 altitudes just above 6000 m, is added**
**to the HSRL-2 AOD at ~50 m above the CTH.**






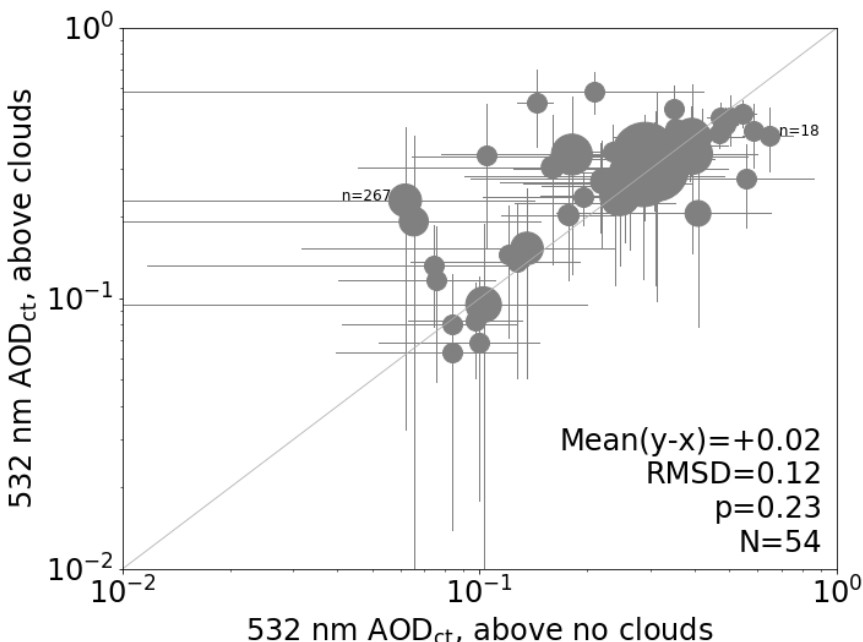


**Figure 4. The meso-scale monthly-mean samples of the AOD above cloud top height.
Each marker represents the mean over a box shown in Fig. 2. The bar represents the
±1 standard deviation range. The marker size is proportional to the number (n) of 10
s measurements, the fewer of the cloudy and clear groups.**










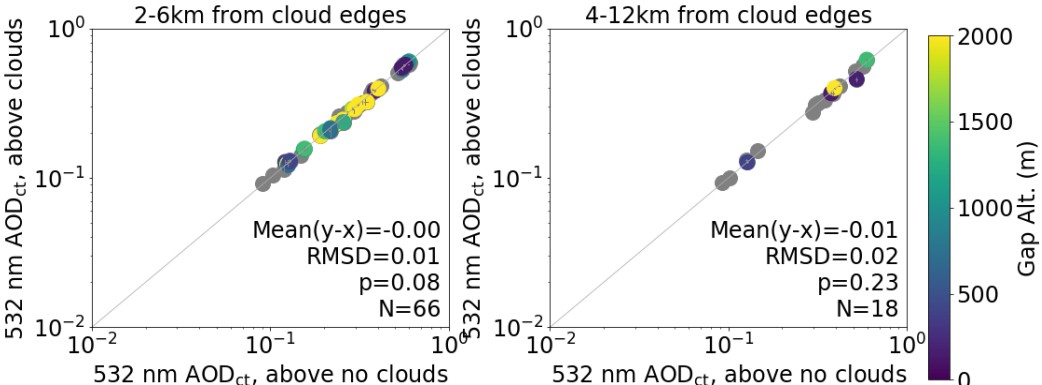

**Figure 5. (a) The local-scale near-synchronous samples of the AOD above cloud top height.**
**Each marker represents the mean over the cloudy and clear sides of a cloud edge, each 2-6**
**km from the edge. The bars indicate the standard deviation of the measurements in each**
**side, almost all of them too short to be discernible. (b) Same as (a) except for the horizontal**
**separation of 4-12 km.**



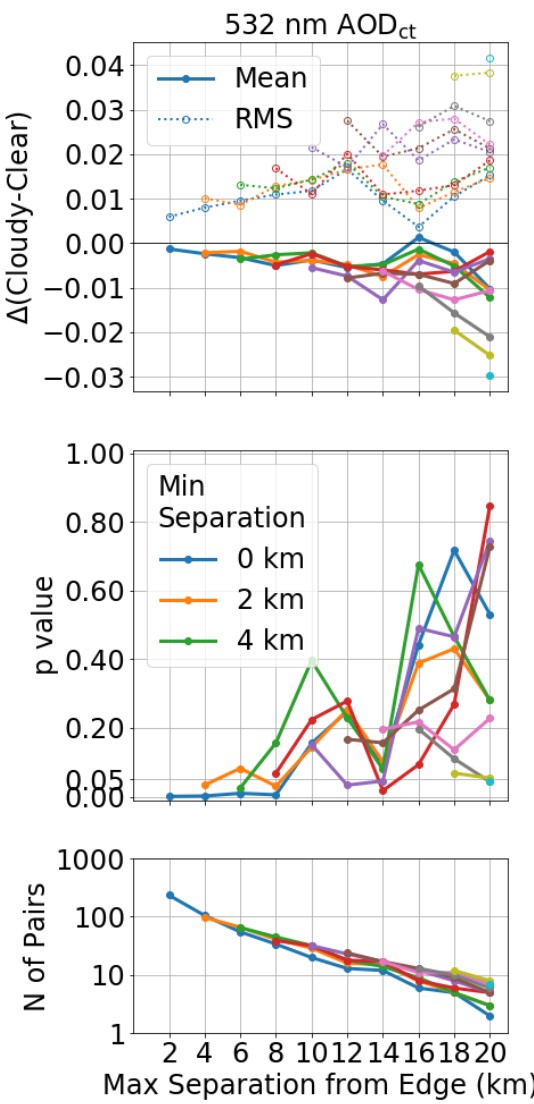


**Figure 6. (a) The mean and root-mean-square deviations of the AOD above cloud top between the cloudy and clear sides of cloud edges. Each side is defined by the horizontal separation from cloud edge. The maximum separation (e.g., 12 km in Fig. 3) is indicated on the x axis. Each line represents the minimum temporal separation (e.g., 4 km in Fig. 3) of 0, 2, 4, …, 18 km in descending order of line length. (b) The p values determined through the paired t-test. (c) the number of cloudy/clear pairs.**