# Peer review of "Daytime aerosol optical depth above low-level"

_Atmospheric Chemistry and Physics, 2019_

## Referee Comment (RC1) · Eric Wilcox (Referee) · 18 Apr 2020

This paper provides a valuable check on a frequent assumption in aerosol cloud studies: that aerosol properties, including optical thickness and intrinsic properties, such as particle optical properties, in cloudy skies are similar to those same properties in the clear-sky areas between clouds. This assumption is made in a large segment of the literature using passive remote sensing data of aerosols and clouds to investigate possible effects of aerosols on clouds, such as microphysical or radiative effects. In particular, the paper focuses on the southeast Atlantic Ocean case where light-

absorbing smoke resides above stratocumulus clouds. In doing so, they also provide a more detailed commentary on our previous paper (Chung et al. 2016), which found a curious difference in aerosol optical thickness (AOT) above cloud compared to clear air near clouds in CALIOP data that is different for daytime overpasses than for nighttime overpasses. This new paper is a useful application of recent high-quality airborne measurements to revisit these issues and is a valuable contribution. I recommend some revisions based on the following comments.

While the authors find no statistically significant difference in AOT above cloud compared to the adjacent cloud-free region, they do find a significant difference in the particle number concentration under the mesoscale averaging method, which combines observations from within 2 degree lat/lon boxes. This result is at odds with the finer scale analysis of the same quantity. To explain, the authors refer to a criterion on "outliers", but I think there is room for more discussion. In addition to clarifying what the outlier criterion is, I wonder if the extreme variability of this quantity in airborne measurements of the free troposphere due to layering of aerosol plumes, for example, make it likely that the sampling is insufficient to even address this metric. One way to evaluate this for particle concentration, as well as the other metrics, is to ask the question: given the observed variability is it likely that the number of samples present is sufficient to adequately sample clear and cloudy air within the 2 deg. box at all? Given that analysis of cloudy air to adjacent clear air shows no significant difference, I suspect that the real problem here is that you have not adequately measured the average particle concentration in the box. This question should probably be evaluated for particle concentration and the other metrics under the meso-scale sampling method.

On the matter of reconciling these new results with the previous Chung et al. (2016) paper, I recommend clarifying a couple of points:

(1) One of the pieces of evidence that the differences between above cloud and clear-air AOT might be an artifact of the CALIOP data is the contrast between the daytime and nighttime difference. This is emphasized by Chung et al. where they note that "The

global average AODct above low clouds is much larger during the nighttime than during the daytime for all seasons (Table 1), which is most likely related to a higher S/N [signal-to-noise ratio] for nighttime retrievals compared to those during the daytime" (p. 5788) and again on p. 5790. While the submitted manuscript mentions that detecting aerosol layers is sensitive to the signal-to-noise ratio which is impacted by the background lighting conditions (lines 67 and 304), it does not clearly make the point that this leads to a strong day/night contrast in the detectability limit. This is, of course, well known in the remote sensing community, but I think that the paper could make this part of the work more accessible to a broader readership by expressing this simple point more directly.

(2) The paper uses data from HSRL and finds no evidence of the differences between above-cloud and clear-air AOT reported in the Chung et al (2016) paper, and builds further support that the differences reported in Chung et al. (2016) are an artifact of the CALIOP data. However, on line 73 the authors also note that Kacenelenbogen et al. (2014) "saw no clear bias" in above cloud AOD between CALIOP and HSRL. How are we to reconcile these to points that seem contradictory?

---

## Referee Comment (RC2) · Anonymous Referee #2 · 30 Apr 2020

This paper uses recent airborne high-spectral-resolution lidar (HSRL) and sunphotometer observations to examine an assumption that aerosol properties over low clouds are similar to that in the neighboring clear sky. This assumption is widely used in the studies that investigate aerosol radiative effects in cloudy and clear-sky conditions using passive satellite remote sensing data. The paper referenced a previous study by Chung et al that used the satellite-borne CALIOP observation to examine the aerosol optical depth (AOD) difference in cloudy and clear-sky conditions. The Chung et al paper revealed a large day and night difference that is most likely related to the CALIOP

measurement. CALIOP is an active lidar instrument and can provide globally range-resolved cloud and aerosol vertical profiles. Its return signal is generally weak due to the high altitude (>700 km) of the satellite orbit, and the data SNR for daytime is not as good as that during nighttime in the presence of large sunlight background noise. Therefore, some tenuous aerosol layers can be missed in the feature detection. This can cause underestimate of daytime AOD. The airborne HSRL measurement used in this study has much larger SNR than that of the CALIOP measurement and can provide the aerosol extinction as well as lidar ratio which is modeled in the CALIOP data processing. This paper uses this HSRL dataset to revisit the cloudy and clear-sky AOD difference. The study is very useful and valuable. I suggest the paper published after some revision.

In the paper, the authors concluded that daytime 532 nm AOD over low-level clouds is similar to that in the surrounding clear skies at the same heights. This supports the assumption mentioned above in the geographical region and season investigated. However, some information and analysis results presented in the paper are confusing and need more explanation and clarification. My biggest confusion is with the statistical analysis using t-test in the paper. In figure 6, the mean 532 nm AODct difference approaches zero as the separation distance between the aerosol layer over cloud and in the surrounding clear sky decreases. This is as I expect. However, the p-values from the t-test are smaller than the threshold of 0.05 for smaller separation distances, suggesting that the null hypothesis of zero AODct difference is rejected (refer to lines 227-228) at a confidence level of 95

Some minor comments:

It may be useful to provide the expression of t-test that can help the discussion and interpretation about the results.

The mean difference is compared to RMSD (line 241, line 249 and line 255) in the discussion about the results in this paper. It may make more sense to compare the

mean difference and standard deviation (or RMSD) to the corresponding mean value of each parameter.

In Table 2, statistics of log10532 nm AODct difference are listed, in addition to 532 nm AODct difference. Is there a reason for this? Any additional information can be drawn from it? A little bit more explanation is needed.

——————————————————————

---

## Author Comment (AC1) · 15 Jun 2020

*We would like to thank Dr. Wilcox for the review.*

This paper provides a valuable check on a frequent assumption in aerosol cloud studies: that aerosol properties, including optical thickness and intrinsic properties, such as particle optical properties, in cloudy skies are similar to those same properties in the clear-sky areas between clouds. This assumption is made in a large segment of

the literature using passive remote sensing data of aerosols and clouds to investigate possible effects of aerosols on clouds, such as microphysical or radiative effects. In particular, the paper focuses on the southeast Atlantic Ocean case where light absorbing smoke resides above stratocumulus clouds. In doing so, they also provide a more detailed commentary on our previous paper (Chung et al. 2016), which found a curious difference in aerosol optical thickness (AOT) above cloud compared to clear air near clouds in CALIOP data that is different for daytime overpasses than for nighttime overpasses. This new paper is a useful application of recent high-quality airborne measurements to revisit these issues and is a valuable contribution. I recommend some revisions based on the following comments.

While the authors find no statistically significant difference in AOT above cloud compared to the adjacent cloud-free region, they do find a significant difference in the particle number concentration under the mesoscale averaging method, which combines observations from within 2 degree lat/lon boxes. This result is at odds with the finer scale analysis of the same quantity. To explain, the authors refer to a criterion on "outliers", but I think there is room for more discussion. In addition to clarifying what the outlier criterion is, I wonder if the extreme variability of this quantity in airborne measurements of the free troposphere due to layering of aerosol plumes, for example, make it likely that the sampling is insufficient to even address this metric. One way to evaluate this for particle concentration, as well as the other metrics, is to ask the question: given the observed variability is it likely that the number of samples present is sufficient to adequately sample clear and cloudy air within the 2 deg. box at all? Given that analysis of cloudy air to adjacent clear air shows no significant difference, I suspect that the real problem here is that you have not adequately measured the average particle concentration in the box. This question should probably be evaluated for particle concentration and the other metrics under the meso-scale sampling method.

*We agree. We have modified the sampling method to reduce the error. We exclude the boxes with fewer than 10 samples, as noted now in Section 2.2. Note that the new*

[Figure]

*restriction has reduced the sample data volume only by 5 minutes or less, depending on aerosol property.*

*The particle number concentration no longer shows a low p value, no longer at odds with the fine scale analysis. The original second paragraph of the results section that described the anomaly has been removed. We have updated the first paragraph and Figure 4 for AODct and Table 2 for all other aerosol properties as well. The results section now points out "The only exception [to the high p values] is the organic mass with a p value just under 0.05 (before rounding)."*

*While the modification mitigates the error in representing the boxes, the fundamental limitation remains: The airborne data are not optimized for statistical analysis in the meso-scale monthly-mean. We nonetheless employ this scale to provide the best possible reference point for satellite-based analysis, as stated in Section 2. In parallel, we present the local-scale near-synchronous sampling method to best exploit the spatiotemporal resolution of the airborne data.*

On the matter of reconciling these new results with the previous Chung et al. (2016) paper, I recommend clarifying a couple of points:

(1) One of the pieces of evidence that the differences between above cloud and clearair AOT might be an artifact of the CALIOP data is the contrast between the daytime and nighttime difference. This is emphasized by Chung et al. where they note that "The global average AODct above low clouds is much larger during the nighttime than during the daytime for all seasons (Table 1), which is most likely related to a higher S/N [signal-to-noise ratio] for nighttime retrievals compared to those during the daytime" (p. 5788) and again on p. 5790. While the submitted manuscript mentions that detecting aerosol layers is sensitive to the signal-to-noise ratio which is impacted by the background lighting conditions (lines 67 and 304), it does not clearly make the point that this leads to a strong day/night contrast in the detectability limit. This is, of course, well known in the remote sensing community, but I think that the paper could make this part of the

work more accessible to a broader readership by expressing this simple point more directly.

*We agree. The discussion now says 'As described in Sect. 1, the CALIOP standard algorithm has a detection bias that leads to greater AOD underestimates over clouds than in clear skies due to upward sunlight reflection. The authors [Chung et al. (2016)] emphasize that this bias might explain their results, pointing to a day-night contrast as evidence: "a corresponding difference cannot be seen in the $\Delta AOD_{ct}$ derived from nighttime retrievals [which are free of sunlight reflection]"'. We have chosen the quote on $\Delta AOD_{ct}$ instead of $AOD_{ct}$, to address not only the day-night contrast but also its implications on the cloudy-clear contrast. We have also revised the fourth paragraph of the introduction to emphasize the findings by Chung et al. (2016), with the quote "might simply be a result of systematic differences between the detection thresholds in clear sky and above low bright clouds."*

(2) The paper uses data from HSRL and finds no evidence of the differences between above-cloud and clear-air AOT reported in the Chung et al (2016) paper, and builds further support that the differences reported in Chung et al. (2016) are an artifact of the CALIOP data. However, on line 73 the authors also note that Kacenelenbogen et al. (2014) "saw no clear bias" in above cloud AOD between CALIOP and HSRL. How are we to reconcile these to points that seem contradictory?

*As described in Kacenelenbogen et al. (2014), an error in the daytime CALIOP standard AOD product mainly originates from either (i) a misclassified aerosol type (and hence, a wrongly assumed lidar ratio in the CALIOP algorithm) and/ or (ii) a low SNR (especially when solar light is reflected on the underlying cloud). Factor (i) will either over or underestimate CALIOP AOD whereas factor (ii) will lead to undetected aerosol layers above clouds (i.e. the underestimation of the aerosol geometrical thickness) and an underestimation of the CALIOP AOD.*

*In Chung et al. (2016), the authors compare CALIOP standard AOD products in clear*

*skies and above near-by clouds. The lower daytime CALIOP AOD above clouds can be explained mainly by factor (ii), as there is no reason to assume a different aerosol classification bias (i.e., factor i) in both clear skies and above near-by clouds.*

*Kacenelenbogen et al. (2014) saw no clear bias in above cloud AOD between CALIOP and HSRL due to the fact that both factors (i) and (ii) are at play and also, possibly, that the study is based on low AAC AOD values (limited number of coincident HSRL-CALIOP AAC cases over Northern America with a majority of AAC AOD < 0.1 and average HSRL AAC AOD of 0.04 $\pm$0.05). On the other hand, and similar to the results in Chung et al. (2016), Liu et al. (2015) describe a CALIOP standard daytime AOD underestimation above clouds over two regions of high AAC AOD values (i.e., Saharan dust across the Atlantic and Smoke in the South East Atlantic). In Liu et al., (2015), while both factors (i) and (ii) are also at play, they mainly explain the CALIOP AAC AOD underestimation by factor (ii) in the South East Atlantic, and by factor (i) in the case of Saharan dust (see their Table 2).*

*We have inserted the explanations in the discussion section.*

*We have made several voluntary changes. The buffer below the P-3 is now described as "a certain depth, 1500 m for most flights" instead of 1500 m. Fig. 3c, originally of a 2018 flight, has been replaced with a 2017 flight. The word environment is now treated as countable. A small number of phrases such as "Going back to the present aircraft-based study" have been inserted for a better flow. See also the other review and our response to it.*

**Anonymous Referee 2**

*We would like to thank Anonymous Referee 2 for the review.*

This paper uses recent airborne high-spectral-resolution lidar (HSRL) and sunpho-tometer observations to examine an assumption that aerosol properties over low clouds are similar to that in the neighboring clear sky. This assumption is widely used in the

studies that investigate aerosol radiative effects in cloudy and clear-sky conditions using passive satellite remote sensing data. The paper referenced a previous study by Chung et al that used the satellite-borne CALIOP observation to examine the aerosol optical depth (AOD) difference in cloudy and clear-sky conditions. The Chung et al paper revealed a large day and night difference that is most likely related to the CALIOP measurement. CALIOP is an active lidar instrument and can provide globally range-resolved cloud and aerosol vertical profiles. Its return signal is generally weak due to the high altitude (>700 km) of the satellite orbit, and the data SNR for daytime is not as good as that during nighttime in the presence of large sunlight background noise. Therefore, some tenuous aerosol layers can be missed in the feature detection. This can cause underestimate of daytime AOD. The airborne HSRL measurement used in this study has much larger SNR than that of the CALIOP measurement and can provide the aerosol extinction as well as lidar ratio which is modeled in the CALIOP data processing. This paper uses this HSRL dataset to revisit the cloudy and clear-sky AOD difference. The study is very useful and valuable. I suggest the paper published after some revision.

In the paper, the authors concluded that daytime 532 nm AOD over low-level clouds is similar to that in the surrounding clear skies at the same heights. This supports the assumption mentioned above in the geographical region and season investigated. However, some information and analysis results presented in the paper are confusing and need more explanation and clarification. My biggest confusion is with the statistical analysis using t-test in the paper. In figure 6, the mean 532 nm AODct difference approaches zero as the separation distance between the aerosol layer over cloud and in the surrounding clear sky decreases. This is as I expect. However, the p-values from the t-test are smaller than the threshold of 0.05 for smaller separation distances, suggesting that the null hypothesis of zero AODct difference is rejected (refer to lines 227-228) at a confidence level of 95

*True, the null hypothesis is rejected for this subset of the statistical analysis. The small*

*p-values and large t-values are explained by the small standard error, which in turn results from the small standard deviation and the large sample number.*

*Our discussion, nonetheless, puts little emphasis on this subset of analysis, because of sampling ambiguity and practical insignificance. First, the transition of aerosols to activated droplets in the so-called twilight zone makes the definition of clear and cloudy sides ambiguous. Second, the mean AOD difference as close to zero as -0.002 has little practical significance to climate science, notwithstanding its statistical significance relative to the standard error.*

*The manuscript now clarifies "[...] most of which, with minimum separation of 0-2 km, are subject to potential ambiguity associated with the so-called twilight zone (Sect. 2.2.2).". "Given that a p-value of 0.05 simply means that there is a one in 20 chance that the null hypothesis is correct, we expect some low p-values just by chance as we conduct many comparisons."*

Some minor comments:

It may be useful to provide the expression of t-test that can help the discussion and interpretation about the results.

*A mathematical expression has been inserted in Section 2.3.*

The mean difference is compared to RMSD (line 241, line 249 and line 255) in the discussion about the results in this paper. It may make more sense to compare the mean difference and standard deviation (or RMSD) to the corresponding mean value of each parameter.

*True. Table 2 now has a column for the mean value. The result section and Figure 4 refer to this.*

In Table 2, statistics of log10532 nm AODct difference are listed, in addition to 532 nm AODct difference. Is there a reason for this? Any additional information can be drawn from it? A little bit more explanation is needed.

[Figure]

*Both linear and logarithmic scales are common when plotting AOD. We just wanted to reassure that our conclusions hold regardless of scale. We have added "something we tested just to confirm that our conclusions do not depend on the choice of linear or log scale",*

*We have made several voluntary changes. The buffer below the P-3 is now described as "a certain depth, 1500 m for most flights" instead of 1500 m. Fig. 3c, originally of a 2018 flight, has been replaced with a 2017 flight. The word environment is now treated as countable. A small number of phrases such as "Going back to the present aircraft-based study" have been inserted for a better flow. See also the other review and our response to it.*